# Psychometric Properties of the Italian Version of the Leader Member Exchange Scale (LMX-7): A Validation Study

**DOI:** 10.3390/healthcare11131957

**Published:** 2023-07-06

**Authors:** Marzia Lommi, Ippolito Notarnicola, Rosario Caruso, Laura Iacorossi, Francesca Gambalunga, Laura Sabatino, Roberto Latina, Teresa Rea, Assunta Guillari, Maddalena De Maria, Rocco Mazzotta, Gennaro Rocco, Alessandro Stievano, Raffaella Gualandi, Daniela Tartaglini, Dhurata Ivziku

**Affiliations:** 1Department of Biomedicine and Prevention, University Tor Vergata, 00133 Rome, Italy; marzia.lommi@gmail.com (M.L.); mazzo.rm@gmail.com (R.M.); 2Centre of Excellence for Nursing Scholarship, Order of Nurses of Rome, 00173 Rome, Italy; ippo66@live.com (I.N.); laura.sabatino8@gmail.com (L.S.); genna.rocco@gmail.com (G.R.); 3Clinical Research Service, IRCCS Policlinico San Donato, San Donato Milanese, 20097 Milano, Italy; rosario.caruso@grupposandonato.it; 4Department of Biomedical Sciences for Health, University of Milan, 20133 Milano, Italy; 5Nursing Research Unit IFO, IRCCS Regina Elena, National Cancer Institute, 00144 Rome, Italy; laura.iacorossi@gmail.com; 6Department of Health Professions (DAPS), University Hospital “Policlinico Umberto I”, 00161 Rome, Italy; francescagambalunga@86gmail.com; 7Department of Health Promotion, Mother and Child Care, Internal Medicine and Medical Specialities, University of Palermo, 90128 Palermo, Italy; roberto.latina@unipa.it; 8Public Health Department, University of Naples Federico 2, 80131 Naples, Italy; teresa.rea@unina.it; 9Department of Nursing, Federico 2 Hospital, 80131 Naples, Italy; assunta.guillari@unina.it; 10Department of Health Professions, Catholic University “Our Lady of Good Counsel”, 1000 Tirana, Albania; 11Department of Clinical and Experimental Medicine, University of Messina, 98100 Messina, Italy; alessandro.stievano@gmail.com; 12Department of Health Professions, Fondazione Policlinico Universitario Campus Bio-Medico, 00128 Rome, Italy; r.gualandi@policlinicocampus.it (R.G.); d.tartaglini@policlinicocampus.it (D.T.); 13Vice President Italian Society for the Direction and Management in Nursing (SIDMI), 00198 Rome, Italy

**Keywords:** LMX-7, leader–member exchange relationships, leadership, measurement, instrument, psychometric, reliability and validity, nurses, nurse manager

## Abstract

For decades, scholars have studied leader–member exchange (LMX) relationships to understand and explain the effects of leadership on follower attitudes and performance outcomes within work settings. One available instrument to measure these aspects is the LMX-7 scale. This measurement has been widely used in empirical studies, but its psychometric properties have been poorly explored. The aim of this study was to test the psychometric characteristics (content, structural and construct validity, and reliability) of the Italian version of the LMX-7 scale and to support its cultural adaptation. We used a cross-sectional multi-center design. The forward–backward translation process was used to develop the Italian version of the scale. The scale was administered through an online survey to 837 nurses and nurse managers working in different settings. The factorial structure was tested using both exploratory and confirmatory factor analyses (EFA and CFA), and reliability was evaluated using Cronbach’s alpha. For the construct validity, we used hypothesis testing and differentiation by known groups. The Italian version of the LMX-7 scale presented one dimension. All the psychometric tests performed confirmed its validity and suggested its usefulness for future research.

## 1. Introduction

As human beings, we are inherently social creatures, and our interactions with others form the basis for the relationships we develop. Relationships are particularly important in the workplace as the quality of social relationships significantly influences the culture, climate, productivity, and success of the organization as a whole [1,2]. Relationships are also essential in leadership and management. Nurturing and satisfying the relational needs of followers is crucial for effective leadership. Leaders who do this well are better at leading their organizations and keeping members engaged [1].

For more than four decades, researchers have studied leader–member exchange (LMX) relationships to understand and explain the effects of leadership on follower attitudes and performance outcomes within work settings [3]. The social exchange theory, role theory, relational leadership theory, and LMX construct have dominated research on organizations and management [4]. Despite the extensive body of literature, and variety in the explored areas, the LMX literature has been criticized for a lack of coherence between theory and empirical research [5]. This is due to the constantly evolving and expanding nature of the construct, making it a complex multi-level and multi-disciplinary field [6,7]. Despite these challenges, LMX continues to be an essential area of study [4,6,8]. Ongoing research will continue to refine our understanding of LMX and its influence on different organizational outcomes, as well as the factors that shape these relationships.

Scholars propose that the quality of the LMX relationships serves as a bridge linking leadership actions and the effectiveness of subordinates [4,8]. Evidence deriving from many meta-analyses indicates that when employees perceive high-quality LMX relationships, they tend to experience improvements in their skills, job performance, empowerment, willingness to get involved in occupational citizenship behaviors, trust in and satisfaction with the leader, job satisfaction, commitment to the organization, and clarity about their roles, while also experiencing reduced turnover intentions [4,7,8]. Additionally, when employees perceive high-quality LMX relationships, negative outcomes such as role conflict and counterproductive performance tend to be reduced [4]. This evidence highlights the importance of evaluating LMX relationships in organizations.

Quality of the leader–member relationship was explored in some studies in healthcare organizations as well. Indeed, high-quality LMX relationships were associated with increased satisfaction with the leader [9], organizational commitment [10,11,12], job satisfaction [11], teamwork satisfaction [13], and reduced turnover intentions among nurses [9,11,14], with some differences between the private and the public healthcare sectors [9,12]. Additionally, social interactions within work contexts can be enhanced by trust, which plays a critical role in facilitating the creation of efficient teams and fostering positive relationships between staff and managers [15]. Moreover, leadership and the quality of human resource management are associated with a reduction in errors and better quality of nursing care [16]. Overall, high-quality relationships and reciprocal trust in work environments are associated with positive outcomes for both individuals and organizations [17]. Therefore, building effective and successful relationships with followers, improving working conditions, creating better work contexts, and enhancing wellbeing are essential to retain and motivate the nursing workforce and to strengthen healthcare organizations’ appeal to nursing professionals [18,19,20,21]. This is particularly important in the current context of a nursing shortage [22].

Experts in the LMX domain have devoted considerable effort to devising measurements that assess the quality of LMX, creating various instruments spanning from one to four dimensions. However, their primary focus has been on empirical research analysis, while psychometric testing of the scales has been largely overlooked [23]. This has resulted in an important shortcoming in LMX research, which has drawn increased criticism of the measurements and construct [5], indicating a gap in critical scale validity analysis. Given the current issues, it is advisable to conduct a more comprehensive exploration of the psychometric proprieties (validity and reliability) of LMX instruments.

The LMX-7 scale, developed by Graen and Uhl-Bien in 1995 [7], is the most frequently used for measuring the quality of a relationship between leaders and followers in terms of mutual respect, reciprocal trust, and expectations of relational obligations. Researchers have employed the LMX-7 to explore leader–follower relationships in various professional settings and cultures [3,5,23]. The LMX-7 scale has been used in various studies conducted on the Italian population. These studies include research conducted among hospital clinical staff [16,24], nurses working in private and public hospitals [9,10,25,26], warehouse personnel [27], police officers [28], furniture manufacturing employees [29], new hire–supervisor future relationship [30], call center workers [31], and retail company employees [32]. The main focus of these studies was to test various predictive, mediation, and moderation models, rather than the psychometric properties of the instrument. In some of these papers, only limited validity testing was present. To the best of our knowledge, there is no evidence on the translation process and robust psychometric testing of the LMX-7 scale in Italian. 

Adapting an instrument culturally to a different language from the original is important. Translation and psychometric validation of instruments is essential for comparing valid and reliable results across different studies and cultures [33]. Using psychometrically tested instruments also ensures that the results are robust and can be replicated in different settings, which is critical for building strong evidence, new theoretical perspectives, and informing clinical practice, education, and policy.

Therefore, the aim of this study was to test the psychometric characteristics (content, structural and construct validity, and reliability) of the Italian version of the LMX-7 scale and to support its cultural adaptation. Having an Italian version of the scale with robust psychometric validation would be important for a comprehensive understanding of the quality of the relationship between leaders and followers in clinical settings.

## 2. Materials and Methods

### 2.1. Design and Theoretical Background

This study has a cross-sectional multi-center design. This research is based on the social exchange theory (SET) [34].

Organizational behavior like LMX can be explained by the conceptual framework of SET. It is based on the premise that effective workplace relationships yield both tangible and intangible positive benefits and outcomes for all stakeholders involved [34]. A fundamental assumption underlying this concept is that relationships, like LMX quality, naturally progress over time towards establishing trust, loyalty, and mutual commitments as an identifying outcome of favorable social exchanges [35]. Empirical research, as described previously, suggests that trust and LMX are important managerial organizational behaviors. LMX and trust act as mediators between leadership behaviors and follower’s work attitude and performance outcomes [4,36]. In the effort to test such relationships in an Italian nurses and nurse managers sample, the research team is performing preliminary secondary analysis of data to test the psychometric properties of some measurements included in the model [15].

### 2.2. Sample and Setting

We were interested in exploring the quality of relationships between nurse managers and registered nurses working in hospitals or in community settings in different regions of Italy. We included various work settings in our exploration of the phenomenon to gain a deeper understanding. 

To be included in the study, registered nurses had to be employed in a public or private healthcare organization, work in teams and collaborate with other nurses or nurse assistants, have worked in the service for at least two months, and be willing to voluntarily participate in the study. 

The exclusion criteria were as follows: working as a freelance registered nurse, working alone in settings where there is no peer team collaboration, being a newcomer in the service or working for less than two months, not being assigned to a stable work setting, being back in the service for less than two months after a long absence, and refusing to participate in the study. The sampling was convenience-based, and participation was voluntary. According to Haier (2016) guidelines, the ideal sample size for CFA analysis for medium models composed of 5 to 9 observed variables is more than 500 participants [37]. In this particular study, all responses from participants enrolled in the parent study at the time of the analysis were included. The final sample size for the CFA analysis consisted of 837 participants.

### 2.3. Data Collection

The principal investigators (DI and DT) promoted the study through the network of the Italian Scientific Society for the Direction and Management of Nursing (SIDMI) by meeting with Nurse Executive Officers who expressed interest. For each study center, a local contact person explained the study to nurse managers and nurses and encouraged participation.

Data were collected from August 2022 to January 2023 using an online survey on the Google Forms platform. The survey included an explanation of the study aims and participation process, followed by an informed consent and data treatment section. Nurses could choose whether to participate in the study and complete the entire survey or only parts of it. All data were collected anonymously.

### 2.4. Instruments

To measure the quality of relationships between nurse managers and nurses, we used the LMX-7 scale [7]. The scale is unidimensional and has 7 items, which are measured using a 5-point Likert scale rated from 1 (never) to 5 (frequently). The overall scores can be understood as very low (7–14), low (15–19), moderate (20–24), high (25–29), and very high (30–35). For this study, the LMX-7 scale was translated into Italian with the permission of the original developers of the scale [7]. The cross-cultural translation, adaptation, and validation of the scale from the original English version to Italian followed the guidelines recommended by Beaton [38] and Sousa and Rojjanasrirat [39].

Two independent bilingual translators, one of whom was a nurse, performed the forward translation of the scale. The translated versions were then combined into a single version by the work group, with minor wording changes made to avoid ambiguity. To ensure cross-cultural equivalence, a group of experts consisting of two researchers, one human resources expert, a psychologist, six nurse managers, and ten nurses was assembled. They evaluated the items for comprehension and comprehensiveness in the Italian language, as well as the relevance of the items with respect to the construct being measured. Following this, back translation was performed by two other bilingual translators, and similarity between the original text and the reverse translations was evaluated by two native English speakers. The Italian version of the scale is present in Appendix A.

Next, nurses were asked to express their satisfaction levels with their role, interdisciplinary team, leader, and organization. We used single items with a 5-point Likert scale ranging from 0 (very unsatisfied) to 4 (very satisfied) to collect this information. A higher score on the Likert scale indicates a higher level of satisfaction.

In addition, nurses were asked about their intention to leave the current work setting, the organization, or the profession altogether. This intention was measured using single items with binary response options: 1 (yes, I intend to leave the service within the next six months) and 2 (no, I do not intend to leave the service). 

Nurses were asked to rate their level of perceived closeness with the nurse manager on a scale from 1 to 10, with 10 indicating a higher degree of perceived closeness.

To measure satisfaction, intention to leave, and closeness to the manager, we used single items purposely developed for the study. This choice is supported by the literature, which suggests using single-item measurements when evaluating a concrete construct [40].

To measure organizational commitment, we used the organizational commitment scale from the Questionnaire on Experience and Work Evaluation (QEEW 2.0 © SKB) developed by van Veldhoven et al. (2015) [41]. The scale comprises six positively worded items and is rated on a 5-point Likert scale ranging from 0 (strongly agree) to 4 (strongly disagree). The scale has solid psychometric properties, with an internal consistency of 0.80, and is already available in Italian. The scoring of the scale ranges from 0 to 100, with lower scores indicating higher levels of organizational commitment.

### 2.5. Data Analysis

The statistical processing of the results was performed using IBM SPSS Statistics, version 26.0 (IBM Corporation, Armonk, NY, USA), the Amos add-on (Armonk, NY, USA).

Descriptive statistics were used to summarize the demographic and sample data analyzed in our study. Continuous data were tested for normality distribution and were presented as the mean and standard deviation (SD). Frequencies and percentages were used to describe ordinal and nominal data. During preliminary analysis missing data, outliers, homogeneity of variance, linearity, and multicollinearity assumptions were examined.

The content validity of the LMX-7 scale was tested following the recommendations proposed by the COnsensus-based Standards for the selection of health Measurement INstruments (COSMIN) and instrument development [42].

A panel of 20 experts, including 2 researchers, 1 human resources expert, a psychologist, 6 nurse managers, and 10 nurses, used the content validity index (CVI) to verify the relevance and clarity of the items. Through a Delphi survey, the experts evaluated the content validity ratio (CVR), the CVI for the scale (S-CVI), and the CVI in the item level element (I-CVI) [43,44]. For the first aspect, the panelists were asked to indicate whether an item was essential or not to make a construct work in a set of items. Each item had a score from 1 to 3: “not necessary”, “useful but not essential”, and “essential”. The CVR varies between 1 and −1. The higher score indicates further agreement of the panelists on the necessity for an item to be included in an instrument. The CVR formula is CVR = (Ne − N/2)/(N/2), where Ne = number of experts voting ‘essential’ and N = total number of recruited experts. The numerical value of the CVR is indicated by the Lawshe table [44]. In our study, CVR was greater than 0.42, thus the items achieved a meaningful level of acceptance [45]. 

Regarding the S-CVI and the I-CVI [43], they range from −1 to +1, and a value of ≥0.70 is considered adequate for keeping the item in the translated version [45,46]. Content validity is generally seen as the initial stage of a complex validation process that often requires a more inferentially robust analysis to determine the construct validity, such as multivariate latent variable modelling (e.g., exploratory factor analysis, confirmatory factor analysis, and model fit).

Exploratory (EFA) and confirmatory factorial analyses (CFA) were performed to test the structural validity of the LMX-7 scale. Before performing the EFA, the suitability of the data for factor analysis was assessed. According to Nunnally, the measure of Kaiser–Meyer–Olkin (KMO) sampling adequacy ranges from 0 to 1 with a value >0.50, and the Bartlett sphericity test should obtain a *p*-value less than 0.05 to be considered suitable to perform EFA [47]. The following criteria were used to determine the number of significant factors: (1) eigenvalues greater than 1.0, (2) Cattell scree plot, (3) the percentage of total variance explained, and (4) elements with loads greater than 0.40 in absolute value. According to Nunnally, factors were estimated using principal component analysis with varimax rotation and maximum likelihood estimation in EFA [47].

For the CFA, we tested the a priori one-dimensional factorial structure proposed by the authors [7], emerged via the EFA, and presented in other previous studies [11,26,29,31,48]. The goodness of fit of the values was interpreted following the literature recommendations [49]. In the EFA and CFA models, we evaluated the following fit indices: χ^2^; mean square error of approximation (RMSEA), where values < 0.08 indicate acceptable model adaptation and <0.05 indicates a good fit of the model; standardized mean square residue (SRMR), where values < 0.05 indicate a good fit of the model; comparative adaptation index (CFI), where values > 0.95 indicate a good fit of the model; the Tucker–Lewis index (TLI), where values > 0.9 indicate a good fit of the model; adjusted fit goodness index (AGFI), where values > 0.9 indicate a good fit of the model; and the index of adjusted adaptation (NFI), where values > 0.9 indicate a good fit of the model [49,50].

To assess the degree of co-variation of items in the scale relative to the total scale score, the Cronbach’s alpha coefficient was measured. This is an evaluation of a scale’s internal consistency. An alpha coefficient of 0.70 can be considered acceptable; however, values between 0.80 and 0.95 are preferred for the psychometric quality of the scales [49].

The construct validity was tested via hypothesis testing and known group differences [51,52]. Specifically, we hypothesized that LMX scores were significantly correlated with the nurse closeness to manager [53,54], age and work tenure [11,31,55], organizational commitment [7,10,11], satisfaction [7,11,13], and intention to leave [7,9,11,14] scores. Construct validity was further verified by posing the hypothesis that LMX scores in public settings were lower than those in private healthcare settings [9,12]. To test the associations with LMX scores, we used Pearson’s product–moment correlation coefficients with a significant *p* value set at <0.05. Correlations of 0.10–0.29 were considered small, 0.30–0.49 as moderate, and >0.50 as strong [56]. Differences between scores were identified through the *t*-test.

### 2.6. Ethical Considerations

This study was carried out in accordance with the ethical standards and principles outlined in the Helsinki Declaration [57] and was approved by the local ethics committee. The researchers asked for permission from the Board of Directors of each participating center before the administration of the questionnaire. All participants received adequate information regarding the study and were afterwards asked to sign the informed consent form. Data access was restricted solely to the research team.

## 3. Results

### Sample Description

The overall sample consisted of 837 participants (681 nurses and 156 nursing managers). The sample was 77.5% female (*n* = 632), with a mean age of 44.3 (SD ± 11.1) years. The survey collected socio-demographic data (as age, gender, highest level of education) and job-related information (as work position, work experience, and work setting information, such as region, city, organization, and service).

Table 1 shows in detail the socio-demographic characteristics of the analyzed sample.

Of the 20 panel members who participated in this phase, 70% (*n* = 14) were women and their average age was 40.7 years, ranging from 30 to 62 years. The validity indices of the measurements obtained from the evaluation of panel members ranged from 0.84 to 1 for I-CVI and were 0.88 for S-CVI. The interpretation of I-CVI and S-CVI indicated satisfactory indices; I-CVI and S-CVI had scores above 0.70.

The KMO measure of sampling adequacy was 0.898 and the significance of the Bartlett sphericity test was less than 0.001, meaning that EFA can be applied to the dataset obtained [58].

A total of one factor was extracted and rotated, and the cumulative variance explained was 63.84%. All elements of the scale had a load factor ranging from 0.676 to 0.849 (Table 2). 

The results of the confirmatory model produced values of χ^2^ = 25.574, with nine degrees of freedom (df), a χ^2^/df ratio of 3.064, and *p* = 0.001. The CFI was 0.995 and the Tucker–Lewis Index (TLI) was 0.987; NFI was 0.992. The RMSEA value was 0.050 and the SRMR value was 0.017. The one-factor fit model of the CFA is shown in Figure 1.

One of the most common estimates of internal consistency is Cronbach’s α. The LMX-7 scale showed a good internal consistency for each item of the scale. The value of the Cronbach Alpha ranged from α = 0.882 to α = 0.905. The scale had a mean Cronbach value of α = 0.904 (Table 3).

The testing of the construct validity of the LMX-7 was confirmed by most of the hypotheses tested (Table 4). No relationship between LMX scores and the age and tenure of the participants was found. Also, in the known group differences testing in our sample, the LMX-7 scores were higher in the public setting than in the private healthcare companies, confirming a contrary hypothesis.

## 4. Discussion

In this study, we have conducted cultural adaptation and psychometric validation of the LMX-7 scale on an Italian sample. Through rigorous methodological procedures, we have examined the adequacy of a new dataset to the LMX-7 factor model proposed in a previous validation study [59]. 

We used EFA to extract the new factorial structure and CFA to test the adaptation of the one-factor model obtained. The one-factor model exhibited good fit indices [49]. These indices confirmed that the LMX-7 scale maintained its unidimensional factorial structure on the Italian sample. We also confirm that all items exhibited acceptable or good internal consistency with a high Cronbach’s α, indicating high reliability. 

We assessed the construct validity by testing different hypotheses based on theoretical concepts and empirical studies. We confirmed positive relationships between LMX and nurses’ satisfaction with their role, teamwork, leader and organization, organizational commitment, and intention to stay in the setting, organization, and profession [7,10,11,13,14]. We observed that age, gender, and work tenure did not have a significant association with LMX relationship, which is consistent with earlier empirical studies [11,31]. However, our findings contrast with previous research involving Italian healthcare professionals [9] as we discovered that nurses working in public organizations (such as hospitals or community settings) perceived a higher-quality LMX relationship than those employed in the private sector. This trend has also been observed in the UK and Australia [9]. 

The results of our study show that the Italian adaptation of the LMX-7 scale exhibits good psychometric properties, suggesting that the scale can be used with confidence to investigate leader–member relationships between nurses and nursing managers in Italy, which may have important implications for clinical practice and future research.

The theoretical construct of the LMX-7 scale has been conceptualized over the years, with the model suggesting that leaders develop unique relationships with each employee, and the quality of this relationship influences various aspects of followers’ behavior and actions [60]. Despite some previous criticism, our study has confirmed the one-dimensionality of the LMX-7 scale and its psychometric properties, which supports the original theoretical construct [7]. We have also established relationships between the LMX-7 scale and followers’ attitudes and behaviors in the workplace. 

The availability of a well-validated instrument with robust psychometric properties to measure leader–member relationships enables its widespread use in clinical practice, education, and research. The LMX-7 scale can be utilized by middle and top managers to investigate the impact of leadership on staff nurses or to measure the outcomes of specific leadership styles and human resource management training. Additionally, it can be employed in empirical research to explore dyadic or multilevel dynamics between leaders and followers and their impact on leadership outcomes.

### Strengths and Limitations

The strengths of the study are connected to the large sample size and the variety of participants included. Indeed, we recruited nurses from various settings such as community care, hospitals, and private and public organizations, and this allowed us to complete the psychometric analysis with construct validity analysis as well. The absence of missing data is an added strength. 

However, there are still some limitations to the study. The convenience sampling method does not allow us to generalize the findings to all nurses in Italy or other countries. Moreover, the cross-sectional design of the study does not allow us to assess the stability of the scale over time. 

Future research should focus on cross-cultural invariance testing of the LMX-7 scale in diverse samples of nurses from various countries and measurement of scale stability using longitudinal designs.

## 5. Conclusions

The LMX construct and measurements have received increased criticism from scholars regarding scale validity analysis. This study contributes to the literature by evaluating the psychometric properties of the LMX-7 scale. The study confirms the one-dimensional structure of the LMX-7 scale, as confirmed by the EFA and CFA. Additionally, the scale demonstrates good psychometric properties in reliability testing and construct validity. The Italian version of the LMX-7 scale is suitable for use by clinicians, leaders, and healthcare researchers in future empirical studies in Italy. The scale can be further tested in future research to explore additional psychometric properties, such as criterion validity and stability, or for validation in longitudinal studies.

## Figures and Tables

**Figure 1 healthcare-11-01957-f001:**
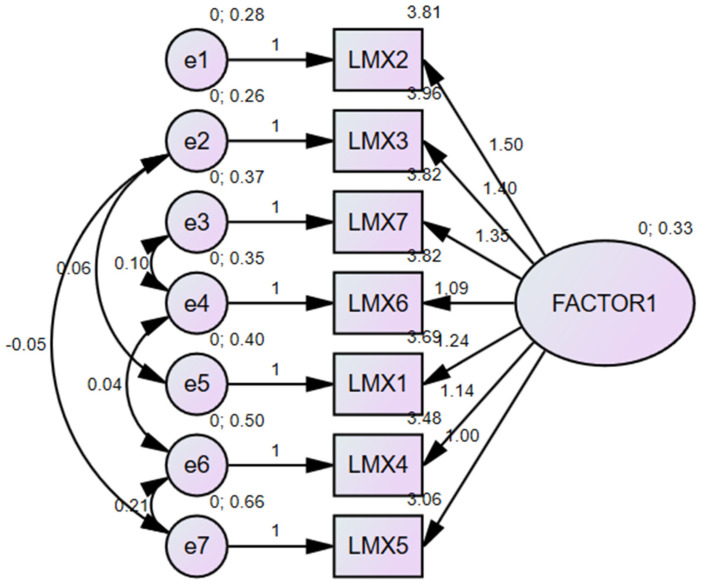
Model of the CFA performed on the LMX-7 scale.

**Table 1 healthcare-11-01957-t001:** Sample characteristics (N = 837 nurses and nurse managers).

	Nurses (*n* = 681)		Nurse Managers (*n* = 156)	
	N	%	N	%
**Region**				
Northern regions	310	45.5	55	55
Central regions	160	23.5	25	25
Southern regions	211	31.0	76	76
**Work setting**				
Public hospital	479	70.3	117	75.0
Public community care	91	13.4	22	14.1
Private hospital	111	16.3	17	10.9
**Age**				
Years (mean; SD)	44.7	11.2	52.35	6.65
**Sex**				
Female	508	75	124	79.5
**Work experience**				
Years (mean; SD)	23.9	19.4	30.42	8.69
**Educational background**				
BSc or equivalent title	180	26.4	7	4.5
Postgraduate certificate after BSc	286	42.0	2	1.3
Master of Science	161	23.6	90	57.7
Postgraduate certificate after MSc	44	6.5	30	19.2
Other postgraduate education	9	1.3	26	16.7
PhD	1	0.1	1	0.6
**Intention to leave the ward/service**				
Yes	183	26.9	27	17.3
**Intention to leave the company/hospital**				
Yes	154	22.6	23	14.7
**Intention to leave the nursing profession**				
Yes	119	17.5	19	12.2
**Satisfaction regarding the current role**				
Score [0 = completely not satisfied; 4 = completely satisfied] (median; IQR)	3	2–3	3	2–4
**Satisfaction regarding multidisciplinary team work**				
Score [0 = completely not satisfied; 4 = completely satisfied] (median; IQR)	3	2–3	3	3–3
**Satisfaction regarding the leadership**				
Score [0 = completely not satisfied; 4 = completely satisfied] (median; IQR)	3	2–4	3	2–3
**Satisfaction with the company/hospital**				
Score [0 = completely not satisfied; 4 = completely satisfied] (median; IQR)	3	2–3	3	2–3
**Closeness to the leader (only on nurses)**				
Score [0 = not at all; 10 = a lot] (mean; SD)	7.5	2.2	-	-
**Organizational commitment scale**				
Score [0 = highest commitment; 100 = lowest commitment] (mean; SD)	49.2	21.7	39.7	20.8

Legend: SD = standard deviation; BSc = Bachelor of Sciences (in Nursing); MSc = Master of Sciences (in Nursing); IQR = interquartile range.

**Table 2 healthcare-11-01957-t002:** Factor loadings results from exploratory factor analysis (N = 871).

Item Scale	Factor 1
LMX-7 1	0.849
LMX-7 2	0.849
LMX-7 3	0.837
LMX-7 4	0.806
LMX-7 5	0.788
LMX-7 6	0.773
LMX-7 7	0.676

**Table 3 healthcare-11-01957-t003:** Internal consistency of scale LMX-7 (N = 871).

	Mean	SD	Alpha of Cronbach	Total
LMX-7 1	3.69	0.954	0.891	0.904
LMX-7 2	3.81	1.011	0.882	
LMX-7 3	3.96	0.954	0.883	
LMX-7 4	3.48	0.968	0.892	
LMX-7 5	3.06	0.996	0.905	
LMX-7 6	3.82	0.863	0.889	
LMX-7 7	3.82	0.990	0.884	

**Table 4 healthcare-11-01957-t004:** Results of the construct validity testing (N = 871).

Hypothesis Testing	Lmx-7	Expected Direction of the Correlation
Closeness to leader	0.318 **	Positive
Age	0.033	Negative
Work experience (tenure)	0.038	Negative
Organizational commitment (reverse score)	−0.312 **	Positive
Satisfaction with teamwork (multidisciplinary)	0.330 **	Positive
Satisfaction with teamwork (peers)	0.254 **	Positive
Satisfaction with job (role)	0.266 **	Positive
Satisfaction with the leader	0.444 **	Positive
Intention to leave organization (reverse score)	0.220 **	Negative
Intention to leave work setting (reverse score)	0.157 **	Negative
Intention to leave profession (reverse score)	0.130 **	Negative
**Differentiation by Known Group**	***t*-Test**	**Expected Difference**
Public versus private settings	5.517 ***	Higher scores in private settings

Notes: Values refers to Pearson correlation coefficients; ** Correlation is significant at the 0.01 level (2-tailed). *** *t* test significant at the 0.001 level (2-tailed).

## Data Availability

The data presented in this study are available on reasonable request from the corresponding author.

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
