# Peer review of "Psychometric Properties of the Italian Version of the Leader Member Exchange Scale (LMX-7): A Validation Study"

_healthcare, 2023, doi:10.3390/healthcare11131957_

Round 1
Reviewer 1 Report
In the abstract and in the sections Design or Sample and setting please provide the total number of nurses surveyed.
Reviewer 2 Report
Dear Authors
Its study is necessary and very interesting. You have made a major effort in a massive collection of data on a large population, so I congratulate you.
I just have a few suggestions to help improve your manuscript that I will list for you:
In the Introduction, line 80, you state "Given the current issues", I wonder if you could develop this idea so that it is clear what you mean. It would also be necessary to end this section with a clear expression of your objective, as found in the abstract.
The Materials and Method section requires a complete revision of the wording, becomes very extensive and cumbersome, losing interest as it progresses.
- There is a lack of methodological support throughout the section, have you followed an approach developed by yourself or is it an approach already developed by other authors? Cite those who defined this way of checking the psychometric properties of scales.
- I have not been able to find information about how the sample size was calculated or about the number of items of the scale they were testing, both of which are necessary.
- It is confusing that you are checking the psychometric properties of a scale and at the same time you are using others that measure other aspects with which they relate the initial construct. This should be justified in the Introduction section and written in a more understandable way in Material and Method.
- Review lines 178-179 that seem to be disconnected from the rest of the text.
- They should reduce the information of each of the sections of the analysis, focusing only on what is important and simplifying the content.
- I think it would be more appropriate for the Ethical Considerations subsection to be at the end of the Material an Methods section, since where it is breaks the reading.
On the Results section, in many cases the subsections begin with a new methodological clarification, sometimes repeated to that found in Material and Method. This is not standard practice. If it is already on your site you do not have to repeat it and if it is not, of course Results is not your site.
Both the Discussion section and the Conclusions have to be improved by relating them to their objective and to the results of their measurement, not distracting by aspects that were not their objective. For example, lines 361-371 are methodological justifications again that do not provide a discursive framework to his work, and from 390-419 we find some continuous turns on the same idea that do not quite work. Finally, I also wonder if this study has not had any limitations.
I insist on congratulating you on your work and I hope that you will address the questions I raise with the intention of improving your manuscript.
Best regards
Reviewer 3 Report
Thank you for having the chance to review this interesting study. I hope this study will make a contribution toward the improvement of nursing management.
I think that this manuscript needs revision. Please check out following.
Keywords: I think that the keywords are drawn from the MeSH list. But keywords except “leadership” and “nursing interns” were not from MeSH list.
1. Introduction: There are repetitions from lines 101 to 111. I think you had better avoid repetition.
2.5. Ethical considerations: I think you had better write the part about ethical considerations more concretely.
2.6. Statistical analysis: After you present the full name (abbreviation) of a word, you should present the abbreviation, not the full name. Some sentences need to be corrected.
You wrote “The content validity ratio formula is CVR = (N e - N / 2) / (N / 2), where N is the total number of panelists.” You should present the meaning of “Ne”.
3.3. Structural validity: Some sentences are already written in the part about the methods. I think you had better avoid repetition. You should correct “0.000” to “0.001” in line 323.
4. Discussion: Some sentences are already written in the part about the methods and results. I think you had better avoid repetition.
Round 2
Reviewer 2 Report
Congratulations!